# Outcome Evaluation on Impact of the Nutrition Intervention among Adolescents: A Feasibility, Randomised Control Study from Myheart Beat (Malaysian Health and Adolescents Longitudinal Research Team—Behavioural Epidemiology and Trial)

**DOI:** 10.3390/nu14132733

**Published:** 2022-06-30

**Authors:** Hazreen Abdul Majid, Ai Kah Ng, Maznah Dahlui, Shooka Mohammadi, Mohd Nahar Azmi bin Mohamed, Tin Tin Su, Muhammad Yazid Jalaludin

**Affiliations:** 1Centre for Population Health, Department of Social and Preventive Medicine, Faculty of Medicine, University Malaya, Kuala Lumpur 50603, Malaysia; maznahd@ummc.edu.my (M.D.); shooka.mohammadi@gmail.com (S.M.); 2Department of Nutrition, Faculty of Public Health, Universitas Airlangga, Surabaya 60115, Indonesia; 3Division of Nutrition and Dietetics, School of Health Sciences, International Medical University, Bukit Jalil, Kuala Lumpur 57000, Malaysia; ngaikah12@gmail.com; 4Department of Sports Medicine, Faculty of Medicine, University Malaya, Kuala Lumpur 50603, Malaysia; naharazmi@um.edu.my; 5South East Asia Community Observatory (SEACO) & Global Public Health, Jeffrey Cheah School of Medicine & Health Sciences, Monash University, Bandar Sunway 47500, Malaysia; tintin.su@monash.edu; 6Department of Paediatrics, Faculty of Medicine, University Malaya, Kuala Lumpur 50603, Malaysia; yazidj@ummc.edu.my

**Keywords:** dietary intake, adolescents, intervention, feasibility study, randomised control trial

## Abstract

A healthy eating environment in the school setting is crucial to nurture the healthy eating pattern for youth. Thus, it helps to combat the obesity issue. However, the impact of healthy school environment on healthy eating habits among Asian adolescents is scarce and less clear. This clustered randomised-control study has two objectives. The first objective was to evaluate the changes in adolescents’ dietary intake after the interventions for all arms (control; healthy cooking training only; subsidization with healthy cooking training). The second objective was to compare the effect of subsidization with healthy cooking training and healthy cooking training only with the control arm on adolescents’ dietary intakes. This study consisted of 340 secondary school students aged 14 years in rural and urban Malaysia. A total of two arms of intervention and one arm of control were included. Intervention one focused on healthy cooking preparation for the canteen and convenience shop operators. Intervention two included subsidization for fruits and vegetables with a healthy cooking preparation training for the canteen and suggestions on providing healthy options to the convenience shop operators. The outcome measured was changes to dietary intake. It was measured using a three-day dietary history pre- and post-intervention. A paired-*t* test was used to evaluate the outcome of intervention programmes on dietary changes for all arms (control, intervention one and two). An ANCOVA test was used to investigate the effect of providing subsidization and healthy cooking preparation training to the canteen and convenience shop operators on adolescents’ dietary intakes as compared to the control arm. Overall, the reduction in energy and carbohydrates for all arms were observed. Interestingly, fat intake was significantly increased after the four-week intervention programme under healthy cooking intervention but not in the food subsidization group. When comparing between control, healthy cooking training only and subsidization with the healthy cooking training arm, there was no significant changes between arms. A robust intervention to include subsidization of healthy foods for intervention programmes at schools in a larger scale study is needed to confirm this finding.

## 1. Introduction

Obesity is a common risk factor for non-communicable diseases (NCDs) such as diabetes mellitus, cardiovascular disease and cancer [1]. It is an alarming trend and needs to be addressed promptly at a young age.

Evidence has shown that creating a healthy food environment at school is crucial to preventing obesity and overweight among adolescents [2]. This leads to the facilitation of making healthier food choices and developing healthier eating patterns among adolescents. Generally, adolescents spend a substantial amount of time at school, having at least one meal there daily [3]. This indicated that the meals taken at the school canteen provide at least a meal of the total daily dietary intake among adolescents. Hence, foods provided by the school canteen play an essential role in combating obesity and improving the overall nutrition status of the students [3,4].

Studies have shown that the availability of healthy foods [5,6,7] and food prices [8] influence the healthy food choices of students in the school canteen. For instance, a study conducted in the United States found that most students preferred to purchase sweets, snacks, and sweet beverages and only consumed one vegetable and fruit per day on average [7]. Furthermore, a local study found that the most served food groups were carbohydrate-based sources (72.5%), whilst the vegetable group was found to be have most limited availability (0.7%). In addition, it was found that one-third of the food served was high in fat [9]. A study found that when healthier food or snacks are sold at a higher price, it leads students to consume unhealthy foods or snacks that are cheaper than the healthier foods [8].

In conclusion, the environment plays a huge role in influencing students’ food decisions. The school food environment must be conducive to facilitate students to develop healthy food choices. Therefore, this study focuses on a four-week intervention programme designed to create a healthy food environment and promote healthy food choices in school cafeterias. With this study, researchers wish to know the feasibility of intervention in the Malaysian schools setting. Therefore, an upscale intervention will be developed based on data collected.

To date, there is increasing evidence on experimental studies among adolescents with regard to creating a healthy school environment in the past decade [10,11,12,13,14]. In general, studies have shown that the positive effects of the intervention helps in combatting obesity issues [10,13,14]. The evidence was often from higher-income countries. However, such similar studies are limited in the Asian region. Therefore, this paper’s first objective was to investigate the changes in adolescents’ dietary intake after the interventions for all arms. The second objective was to assess the effect of subsidization and healthy cooking preparation training to canteen and convenience shop operators on adolescents’ dietary intakes compared to no intervention. It reports outcomes of the interventional part of the MyHeART BEaT (Malaysian Health and Adolescents Longitudinal ResearchTeam Behavioural Epidemiology and Trial) project [15,16,17,18,19].

## 2. Materials and Methods

### 2.1. Overview Study Design and Area

MyHeART BEaT was a pilot, clustered randomised-control study that involved a four-week intervention among secondary school students. A three-armed quasi-experimental approach (control, intervention 1 and intervention 2) was employed. This study covered six secondary schools in Selangor and Perak, Malaysia. The study frame was a complete list of all public schools in the selected regions (Central and Northern). Selangor and the Federal Territory of Kuala Lumpur (FTKL) are situated in the central region and Perak is located in the northern region. The total number of secondary schools was 595 (261 schools were were in Selangor, 238 schools in Perak and 96 schools in the FTKL). This information was retrieved from the MoE Malaysia in 2012. The listed schools were stratified into urban and rural areas based on criteria provided by the Department of Statistics Malaysia before a random sample was performed using a computer-generated random number list. As a result, the sampling frame consisted of six schools originating from original MyHeART in 2012 [20]. Using a computer-generated randomization method, these six schools were randomised into two intervention arms and one control arm within location (urban versus rural). Two schools were assigned to control, intervention one and intervention two, respectively. Intervention one focused on healthy food preparation training for food vendors only, while intervention two focused on healthy food preparation training and in creating a healthy food environment. The control arm continued the regular food service operation without any intervention. The details of this study protocol with summary diagram of dietary intervention were described in a previous paper [15]. The study was registered at the ISRCTN registry with the code: ISRCTN 89649533. The summary of flow of this study is shown in Figure 1.

### 2.2. Study Period

This study was carried out between September 2018 and March 2019. The baseline phase (P0) data was collected between August and September 2018. The intervention phase (Pi) data collection was between January and March 2019.

### 2.3. Study Population

The participants were recruited at the age of 14 years (P0) and participated in the study at 15 years (Pi). The participants were tracked using their identification number, and no additional participants were added during the intervention period. The participants used the baseline phase data; the sociodemographic and anthropometric data, and the outcome of interest (dietary intakes). After four weeks of intervention, all participants were then followed up to record the above-listed data and outcome of interest.

In total, 359 students were recruited during the baseline phase. All data for both phases (P0 and Pi) were completed, and there was no missing data. However, 19 students were excluded because of implausible energy intake (male: <800 kcal/day and >5000 kcal/day; female: <500 kcal/day and >3500 kcal/day) [21]. Thus, the final number of students included for analysis was 340.

### 2.4. Sample Size Calculation

A formal power calculation was not conducted because this study is a feasibility study and was designed to provide initial information on the potential of the intervention approaches [22]. Thus, sample size calculation was not undertaken. Nevertheless, the intervention arms and control arms were grouped based on the minimum recommendation participants for a pilot cluster randomised-control study [23].

## 3. Intervention

### 3.1. Intervention One (Training Only)

Under this intervention arm, training of the canteen and school convenience shop operators was the only focus. Operators were trained on healthy cooking methods, the national healthy eating policy, nutrition, and change management. This arm of intervention is to request canteen and school convenience shop operators to provide healthy foods and drinks and to consider alternative methods of cooking by using a training manual. A training manual was developed by MyHeART BEaT researchers based on the Malaysian Healthy Canteen Guidelines [24]. In addition, they were invited to attend an hour training workshop to educate them on healthy eating policy, nutrition, canteen stock, and change management. During this workshop, they were given a “Healthy Canteen Booklet”.

### 3.2. Intervention Two (Training and Subsidisation)

As for intervention two, creating a healthy eating environment was one of the areas of emphasis besides training canteen and school convenience shop operators. In order to create a healthy eating environment, two strategies (training and subsidization) were employed under this intervention arm. A previously published paper explained the justifications for the chosen strategies [15]. Basically, the aim of intervention two was to increase the (i) intake of fruits and vegetables and (ii) consumption of healthy and lower energy kuihs (kuih apam kukus, kuih ketayap). The subsidy was provided in three forms; (1) the canteen operators received weekly subsidies to sell fruits, vegetables and low energy dense local desserts (kuih apam kukus, kuih ketayap); (2) the studied population received coupons which subsidize the price of fruits and low energy dense desserts weekly; (3) an allocation of funds to prepare healthy food was given to the school. A water fountain was installed to provide free drinking water [15]. This was done to encourage zero calorie drinks among the study participants.

### 3.3. Control

Generally, all schools receive healthy canteen guidelines from the Ministry of Health (MoH) Malaysia. Therefore, the canteen and convenience shop operators continued their usual service under this control arm based on healthy canteen guidelines by MoH. There was no intervention for this control arm.

### 3.4. Outcome Measure

#### Dietary Intake

A three-day diet history was used to assess dietary intake among the adolescents. This method was the most preferred and commonly used method for epidemiological studies [25]. Furthermore, adolescents can recall better than adults using this method [26]. Trained dietitians documented all the food types and amount consumed within a day (breakfast, mid-morning snacks, lunch, afternoon tea, dinner, and supper). A food portion flip chart developed during the exploratory phase was used to estimate portion size [16]. The total energy intake (kcal/d), intakes of carbohydrate (g/d), protein (g/d), fat (g/d), crude fibre (g/d), sugar (g/d) and sodium (mg/d) were analysed using the Nutritionist Pro database (Axxya Systems, Stafford, Texas, TX, USA) software. The nutrient database was based on the Nutrient Composition of Malaysian Food (4th Edition) [27]. Data cleaning was performed to ensure the consistency and correctness of the data entered as well as to identify any implausible energy intake. A total of 19 participants with implausible energy intake were removed from the dietary dataset [21].

### 3.5. Anthropometric Measures

Weight and height were used to calculate body mass index (BMI). Bodyweight was measured using a calibrated, digital electronic weighing scale (Seca 813; Seca, Birmingham, UK). Height was measured with a calibrated vertical stadiometer (Seca Portable 217; Seca). Both measurements were to the nearest 0.1 kg and 0.1 cm, respectively. Waist circumferences was measured with a non-elastic measuring tape (Seca 201; Seca) and recorded to the nearest 0.1 cm. Body fat percentage was measured using a bioelectric impedance analyser (SC−240 Body Composition Analyser; Tanita Europe BV, Amsterdam, The Netherlands).

### 3.6. Sociodemographic Measures

Sociodemographic information on sex, date of birth, ethnicity and place of residency was collected by a questionnaire.

## 4. Statistical Analysis

All the dietary intakes and anthropometry variables were in complete form. Skewness, kurtosis and Kolgomorov–Smirnov tests were conducted to check the normality of data (*n* = 340). Most of the variables were not normally distributed based on the Kolgomorov–Smirnov test. Such a significant result is expected in a large sample [28]. Therefore, the histogram and Q-Q plot were interpreted together with the values for skewness and kurtosis. All of the variables were interpreted as normally distributed except for a few variables for the baseline and intervention phase. It was found that the weight (kg), BMI (kg/m^2^), percentage body fat (%), waist circumference (cm), energy (kcal/d), carbohydrate (g/d) and fat (g/d) intake variables at baseline and weight (kg), BMI (kg/m^2^), hip and waist circumference (cm), energy (kcal/d), carbohydrate (g/d), protein (g/d) and fat (g/d) intakes at the intervention phase were skewed positively.

This study reported continuous data in means and standard deviation, while categorical data were reported in frequencies and percentages for the descriptive statistics. Dietary intake was the primary outcome of interest. Given the potential confounding effect of energy intake, the macronutrients (protein, carbohydrate and fat), sodium, sugar and crude fibre were adjusted accordingly. All nutrient intakes were adjusted using the nutrient density method and expressed as 1000 kcal [21].

The paired *t*-test investigated the effect of the intervention on dietary intake changes for each arm of interventions. The ANCOVA test compared the effectiveness of interventions among all arms (intervention one, intervention two and control). To control the post-intervention effect for the differences in pre-intervention of all outcomes of interest (energy, carbohydrate, protein, fat, sugar, fiber and sodium), baseline measures were entered as the covariate, intervention arms as the independent variable and post-intervention measures as the dependent variable. All effects were evaluated based on a 95% confidence interval (CI) that did not cross zero, which is regarded as statistically significant. P-values were not reported given the feasibility study and the associated lack of statistical power [15]. The statistical analyses were performed using Statistical Package for Social Sciences (SPSS) software for Windows (version 24.0 Chicago, IL, USA).

## 5. Results

### 5.1. Baseline Characteristics of Participants

Table 1 provides the study population’s baseline sociodemographic, anthropometric, and dietary intake data according to intervention arms and control arms.

The mean age of the study population was 14.1 years. Overall, this study has more females (185 students, 60.3%) than males (135 students, 39.7%). The majority was Malay (284 students, 83.5%), followed by other ethnicities (29 students, 8.5%), Indian (18 students, 5.4%) and Chinese (9 students, 2.6%). Half of the participants were residing in a rural area (53.5%).

There were no differences in dietary intakes and anthropometry measures between the intervention arms and the control arm at baseline.

### 5.2. Effects of Interventions on Dietary Intakes and Anthropometry Measures

Table 2 shows the results of the paired-*t* test analysis according to the intervention arms and the control arm. All arms (intervention one, intervention two and control) showed a statistically significant difference in weight (kg), height (cm), energy (kcal/d) and carbohydrate intakes (g/1000 kcal per day). In addition, percentage body fat (%) and fat intakes (g per 1000 kcal per day) were statistically significant differences for intervention one. As for the intervention two arm, waist circumference (cm) had statistically significant differences pre- and post-intervention.

When intervention arms and control arms were compared (refer to Table 3), none of the studied variables showed statistically significant differences between intervention one, intervention two, and control arms.

## 6. Discussion

### 6.1. Effects of Interventions on Dietary Intakes

This study found that there is a reduction in energy and carbohydrate intakes for all arms (control, intervention one and two) after the four-week intervention. All three arms showed a reduction in energy intake ranging between 130–200 kcal per day and carbohydrate intake ranging between 4.8 g to 6.0 g per 1000 kcal per day. This is a good indicator showing that the intervention programmes, regardless of form, can curb the obesity issues to a certain extent [2].

Interestingly, the fat intake increased under intervention one. A possible explanation for this observation could be the availability of high-fat food. Despite intervention one emphasizing that the training was focused on food handlers in providing healthy options, including low-fat cooking, high-fat food was still being sold at the school canteen [9]. A possible explanation is that profit played a role since subsidies were only provided to participants to buy fruits and vegetables. In addition, prepared foods were mainly carbohydrate and protein-based food in which fat was used in the cooking process. Studies have shown that the availability of healthy food provided in the school cafeteria influences the healthy and active choices of the students by an average of 15% [4,11]. Furthermore, the perception of healthy food was a barrier to implementing a healthy school canteen in this study group [18]. The students perceived that food sold at the canteen were generally healthy and suitable for adult consumption [18]. Thus, this could explain that the healthy cooking training for canteen operators may not work well and warrants multiple strategies to improve students’ dietary intake.

When comparisons were made between the control, intervention one and two arms, there was no significant reduction in dietary intakes. Nevertheless, these effects may accumulate over time and potentially become clinically significant if the intervention continues for a longer duration. Numerous studies [29,30,31] and review papers [32,33,34] have shown the small magnitude of effectiveness in improving diets in a school setting. Some of the strategies proven to be successful are policies for the healthy school food environment [34], canteen-based food nutrition education for school children [29], and the provision of fruit and vegetables by canteen providers [33]. Therefore, the importance of robust intervention, including duration, the surrounding community and the personal food environments should be emphasized when a nutrition intervention programme is developed [32,33,34].

### 6.2. Effects of Interventions on Anthropometry Measures

In this study, weight and height were increased after the four-week intervention for all arms (control, intervention one and intervention two arms). Within the control arm, BMI statistically and significantly increased after the four-week intervention. This indicated that the conventional way did not halt the incremental increase in BMI. This is an expected outcome because the adolescents were still growing. This warrants an emphasis on healthy eating by creating a healthy eating environment to promote a healthier bodyweight, especially in terms of lean body mass.

However, we did not observe any significant difference in reduction when a comparison was made between arms (control, intervention one and two). Although positive effects of food subsidization and training on healthy cooking in this study were not observed, the school-based intervention showed promising results in improving the dietary intake among adolescents [19]. Moreover, affordability was associated with better dietary intake [19].

### 6.3. Strengths and Limitations

The randomised controlled trial design of the study has allowed the researchers to investigate the impact of the intervention on dietary intakes among rural and urban areas. This data also gives an insight into the local intervention programme, leading to a healthier eating practice. Besides that, the study used validated methods to assess dietary intakes to measure dietary intakes among adolescents.

Nevertheless, this study has limitations that warrant caution when interpreting the findings. The pilot design means that a bigger sample size is required. However, this pilot study provides fundamental information and understanding in terms of whether such intervention programme should be increased to a larger scale. It is also crucial to engage more stakeholders to ensure the success of the intervention programme to create a healthy eating environment.

## 7. Conclusions

This study showed that intervention programmes have some potential benefits and provide good insights for an upscaling intervention programme at a larger scale. The current study showed a reduction in energy and carbohydrate intakes after a four-week intervention for all arms. Although the magnitude of reduction was small and insignificant at this time, this could potentially have an accumulative effect on reducing these dietary intakes under a more extended intervention period. Nevertheless, these findings emphasize the need to develop a multiple approach to nutritional intervention which includes food subsidies and appropriate duration, and the involvement of more key stakeholders such as Ministry of Health (MoH), Ministry of Education (MoE), Parent-teacher association (PTA), school principals, food vendors, food suppliers and non-governmental organisations (NGOs) for the benefits of the schoolchildren.

## Figures and Tables

**Figure 1 nutrients-14-02733-f001:**
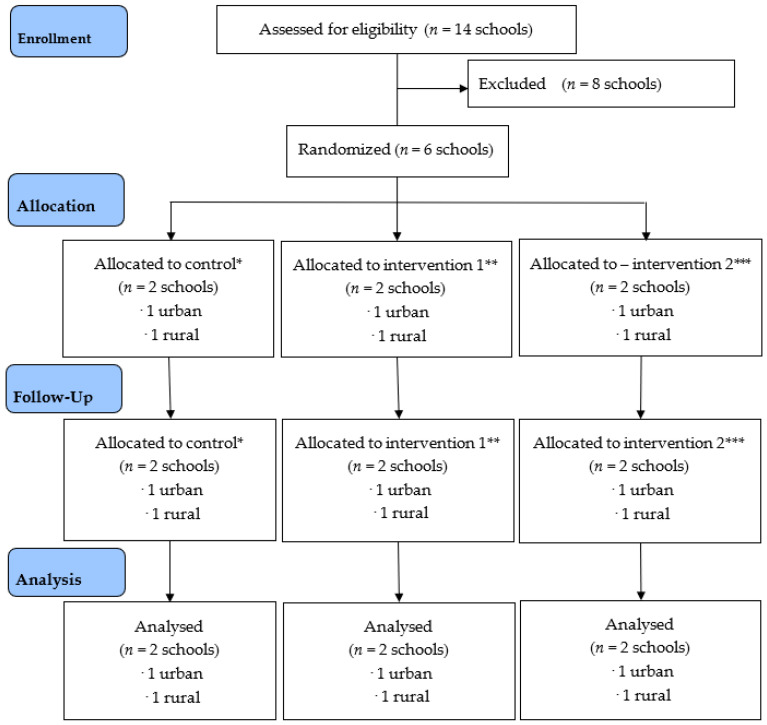
The CONSORT Flow diagram of MyHeART BEaT study. * Control: No intervention. The canteen and convenience shop operators continued their usual approach based on healthy canteen guidelines by MoH. ** Intervention 1: Training only. Training on healthy cooking methods, the national healthy eating policy, nutrition, and change management of the canteen and school convenience shop operators was the only focus. *** Intervention 2: Training and subsidisation. Furthermore, training on healthy cooking methods, the national healthy eating policy, nutrition, and change management, and subsidisation was given to canteen and school convenience shop operators.

**Table 1 nutrients-14-02733-t001:** Baseline demographic, anthropometry and dietary intake of MyHeARTBeAT respondents (mean [SD] or frequencies [%]).

	Intervention 1 [Training Only] (*n* = 136)		Intervention 2 [Training + Subsidy] (*n* = 111)		Control (*n* = 93)		All (*n* = 340)	
Gender	*n*	%	*n*	%	*n*	%	*n*	%
Male	65	47.8	34	30.6	36	38.7	135	39.7
Female	71	52.2	77	69.4	37	61.3	185	60.3
Place of residency	*n*	%	*n*	%	*n*	%	*n*	%
Urban	65	47.8	47	42.3	46	49.5	158	46.5
Rural	71	52.2	64	57.7	47	50.5	182	53.5
Ethnicity	*n*	%	*n*	%	*n*	%	*n*	%
Malay	93	68.4	103	92.8	88	94.6	284	83.5
Chinese	4	3.0	5	4.5	0	0	9	2.6
Indian	15	11.0	3	2.7	0	0	18	5.4
Others	24	17.6	0	0	5	5.4	29	8.5
	Mean	SD	Mean	SD	Mean	SD	Mean	SD
Age, years	14.1	0.4	14.2	0.3	14.1	0.3	14.1	0.3
Weight (kg)	52.4	15.2	50.7	14.6	53.2	15.1	52.1	15.0
Height (cm)	155.7	7.0	155.5	7.0	155.5	7.5	155.6	7.1
BMI (kg/m^2^)	21.5	5.5	20.8	5.1	21.8	5.2	21.3	5.3
Energy and nutrients intake								
Energy (kcal/d)	1991	569	2062	587	2094	568	2042	573
Carbohydrate (g/d)	268.4	91.7	277.6	89.7	284.7	85.4	275.9	89.4
Protein (g/d)	70.7	20.6	75.7	23.4	73.0	21.2	73.0	21.7
Fat (g/d)	70.7	23.3	73.4	25.4	75.3	26.7	72.8	24.9
Sugar (g/d)	47.0	31.0	52.4	36.7	55.5	28.4	51.1	32.4
Sodium (mg/d)	2874.7	1160.6	3034.2	1365.3	3023.9	1430.0	2967.6	1304.6
Fiber (g/d)	5.7	7.6	5.4	4.7	5.2	4.4	5.5	6.0

**Table 2 nutrients-14-02733-t002:** Anthropometry and dietary intake measures at baseline and follow up of MyHeARTBeAT respondents.

	Intervention 1 [Training Only] (*n* = 136)				Intervention 2 [Training + Subsidy] (*n* = 111)				Control (*n* = 93)			
	Mean	SD	Mean Differences	95% CI	Mean	SD	Mean Differences	95% CI	Mean	SD	Mean Differences	95% CI
Weight (kg)												
Baseline	52.4	15.2	1.3 *	0.9 to 1.7	50.7	14.6	1.9 *	1.5 to 2.2	53.2	15.1	1.8 *	1.2 to 2.4
4 weeks	53.7	15.3			52.5	15.1			54.9	16.1		
Height (cm)												
Baseline	155.7	7.0	1.9 *	1.6 to 2.2	155.5	7.0	1.7 *	0.7 to 2.6	155.5	7.5	1.4 *	1.0 to 1.7
4 weeks	157.6	7.1			157.1	8.3			156.9	7.6		
BMI (kg/m^2^)												
Baseline	21.5	5.5	0.0	−0.1 to 0.2	20.8	5.1	0.3	0 to 0.6	21.8	5.2	0.3 *	0.1 to 0.6
4 weeks	21.5	5.5			21.1	5.2			22.1	5.5		
Energy and macronutrients intake												
Energy (kcal/d)												
Baseline	1991	569	130 *	−254 to −7	2062	587	−148 *	−276 to −20	2094	568	−201 *	−324 to −77
4 weeks	1860	600			1915	570			1893	539		
Carbohydrate (g/1000 kcal per day)												
Baseline	133.8	17.3	−4.8 *	−8.1 to −1.5	134.1	16.6	−6.0 *	−10.0 to −2.1	135.9	14.5	−5.4 *	−9.1 to −1.6
4 weeks	129.1	16.9			128.1	16.8			130.5	13.7		
Protein (g/1000 kcal per day)												
Baseline	35.9	5.9	−0.4	−2.1 to 1.2	36.9	6.7	−0.6	−0.9 to 2.1	35.0	5.0	1.8	−0.1 to 3.4
4 weeks	35.5	7.6			37.5	6.2			36.8	6.1		
Fat (g/1000 kcal per day)												
Baseline	35.7	6.8	2.0 *	0.6 to 3.4	35.8	8.6	1.4	−0.5 to 3.3	35.8	8.2	0.7	−1.1 to 2.4
4 weeks	37.8	6.7			37.2	6.6			36.5	4.9		
Sugar (g/1000 kcal per day)												
Baseline	23.6	14.6	2.3	−0.6 to 5.2	24.9	16.4	−0.9	−4.4 to 2.6	26.7	12.6	−0.9	−3.9 to 2.1
4 weeks	25.9	13.4			24.1	11.7			25.8	11.2		
Sodium (mg/1000 kcal per day)												
Baseline	1456.8	445.0	−63.8	−166.6 to 39.0	1483.5	570.0	−51.8	−195.0 to 91.5	1439.0	496.9	22.4	−111.0 to 155.8
4 weeks	1392.9	487.9			1431.7	694.5			1461.3	450.5		
Fiber (g/1000 kcal per day)												
Baseline	3.0	4.3	0.0	−0.92 to 0.93	2.6	2.0	−0.2	−0.66 to 0.14	2.5	2.0	−0.2	−0.67 to 0.37
4 weeks	3.0	3.3			2.4	1.4			2.3	1.6		

Mean difference between baseline and 4-week post intervention using paired-*t* test; * statistically significant if 95% CI that did not cross zero.

**Table 3 nutrients-14-02733-t003:** Effects of intervention on anthropometry and dietary intake measures of MyHeART BEaT respondents.

Variable			Mean Difference	Std. Error	95% Confidence Interval	
					Lower Bound	Upper Bound
Weight (kg)	Control	Intervention 1	0.5	0.33	−0.30	1.26
		Intervention 2	−0.2	0.34	−0.97	0.67
	Intervention 1	Control	−0.5	0.33	−1.26	0.30
		Intervention 2	−0.5	0.31	−1.38	0.11
	Intervention 2	Control	0.2	0.34	−0.67	0.97
		Intervention 1	0.6	0.31	−0.11	1.4
Height (cm)	Control	Intervention 1	−0.5	0.44	−1.58	0.53
		Intervention 2	−0.3	0.46	−1.42	0.78
	Intervention 1	Control	0.5	0.44	−0.53	1.58
		Intervention 2	0.2	0.42	−0.80	1.20
	Intervention 2	Control	0.3	0.46	−0.78	1.42
		Intervention 1	−0.2	0.42	−1.20	0.80
BMI (kg/m^2^)	Control	Intervention 1	0.3	0.16	−0.06	0.70
		Intervention 2	0.0	0.16	−0.36	0.43
	Intervention 1	Control	−0.3	0.16	−0.70	0.06
		Intervention 2	−0.3	0.15	−0.64	0.07
	Intervention 2	Control	0.0	0.16	−0.43	0.36
		Intervention 1	0.3	0.15	−0.07	0.64
Energy (kcal/d)	Control	Intervention 1	2	74	−175.94	180.31
		Intervention 2	−31	77	−216.18	155.11
	Intervention 1	Control	−2	74	−180.31	175.94
		Intervention 2	−33	70	−201.84	136.40
	Intervention 2	Control	31	77	−155.11	216.18
		Intervention 1	33	70	−136.40	201.84
Carbohydrate (g/1000 kcal per day)	Control	Intervention 1	0.9	2.09	−6.84	5.63
		Intervention 2	2.0	2.18	−5.86	7.17
	Intervention 1	Control	−0.9	2.09	−5.63	6.84
		Intervention 2	1.0	1.99	−4.67	7.18
	Intervention 2	Control	−2.0	2.18	−7.17	5.86
		Intervention 1	−1.0	1.99	−7.18	4.67
Protein (g/1000 kcal per day)	Control	Intervention 1	1.4	0.91	−0.83	3.54
		Intervention 2	−0.5	0.96	−2.82	1.77
	Intervention 1	Control	−1.4	0.91	−3.54	0.83
		Intervention 2	−1.9	0.87	−3.70	0.20
	Intervention 2	Control	0.5	0.96	−1.77	2.82
		Intervention 1	1.9	0.87	−0.20	3.96
Fat (g/1000 kcal per day)	Control	Intervention 1	−1.3	0.81	−3.28	0.64
		Intervention 2	−0.7	0.85	−2.78	1.31
	Intervention 1	Control	1.3	0.81	−0.64	3.28
		Intervention 2	0.6	0.77	−1.27	2.45
	Intervention 2	Control	0.7	0.85	−1.31	2.78
		Intervention 1	−0.6	0.77	−2.45	1.27
Sugar (g/1000 kcal per day)	Control	Intervention 1	−0.7	1.62	−4.58	3.22
		Intervention 2	1.4	1.69	−2.65	5.48
	Intervention 1	Control	0.7	1.62	−3.22	4.58
		Intervention 2	2.1	1.54	−1.60	5.79
	Intervention 2	Control	−1.4	1.69	−5.48	2.65
		Intervention 1	−2.1	1.54	−5.79	1.60
Sodium (mg/1000 kcal per day)	Control	Intervention 1	72.3	73.35	−104.20	248.77
		Intervention 2	39.3	76.67	−145.15	223.79
	Intervention 1	Control	−72.3	73.35	−248.77	104.20
		Intervention 2	−33.0	69.74	−200.77	134.83
	Intervention 2	Control	−39.3	76.67	−223.79	145.15
		Intervention 1	33.0	69.74	−134.83	200.77
Fiber (g/1000 kcal per day)	Control	Intervention 1	−0.7	0.33	−1.45	0.12
		Intervention 2	0.1	0.34	−0.87	0.77
	Intervention 1	Control	0.7	0.33	−0.12	1.45
		Intervention 2	0.6	0.31	−0.13	1.36
	Intervention 2	Control	0.1	0.34	−0.77	0.87
		Intervention 1	−0.6	0.31	−1.36	0.13

Mean difference between intervention one, two and control using ANCOVA.

## Data Availability

Not applicable.

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
