# Peer review of "Outcome Evaluation on Impact of the Nutrition Intervention among Adolescents: A Feasibility, Randomised Control Study from Myheart Beat (Malaysian Health and Adolescents Longitudinal Research Team—Behavioural Epidemiology and Trial)"

_nutrients, 2022, doi:10.3390/nu14132733_

Round 1

Reviewer 1 Report

This study aims to assess the changes in dietary intake after interventions in adolescents. This manuscript needs to be improved for the publication. Several comments are suggested.

-Abstract does not sufficiently provide the information on this study. Background should be provided. Study objective should be clearly provided. Results can be added in detail rather than description statistical analysis methods used in this study.

-In introduction, a study background needs to be written in detail to support the objective of this study. The aim of this study is not cleared described. The clear and specific aims need to be addressed.

-It would be better to provide a study intervention diagram for the purpose of understanding this study.

-It would be better to provide a summary table of dietary intervention to improve healthy eating behaviors.

-Discussion should be improved. In lines 261-264, authors need to cite more studies as they mentioned numerous studies.

Author Response

Good day.

Thank you.

Reviewer 2 Report

This is a paper on the outcome evaluation of the impact of nutrition interventions. It is well written, and no major problems are detected. Some minor comments are as follows.

L98-100

In the preposition phrases of "between", "and" should be used instead of "to".

Figure 1

Interventions are named differently (minor or major) from the text. 

Table 1

As 8.5% of the students were grouped into the other ethnicities, its row should be shown in Table 1.

Author Response

Good day.

Thank you.

Round 2

Reviewer 1 Report

This manuscript has been improved for the publication.